# GROUND-TRUTH ADVERSARIAL EXAMPLES

## ABSTRACT

The ability to deploy neural networks in real-world, safety-critical systems is severely limited by the presence of *adversarial examples*: slightly perturbed inputs that are misclassified by the network. In recent years, several techniques have been proposed for training networks that are robust to such examples; and each time stronger attacks have been devised, demonstrating the shortcomings of existing defenses. This highlights a key difficulty in designing an effective defense: the inability to assess a network's robustness against future attacks. We propose to address this difficulty through formal verification techniques. We construct *ground truths*: adversarial examples with a provably-minimal distance from a given input point. We demonstrate how ground truths can serve to assess the effectiveness of attack techniques, by comparing the adversarial examples produced by those attacks to the ground truths; and also of defense techniques, by computing the distance to the ground truths before and after the defense is applied, and measuring the improvement. We use this technique to assess recently suggested attack and defense techniques.

## 1 INTRODUCTION

While machine learning, and in particular neural networks, have seen significant success, recent work Szegedy et al. (2014) has shown that an adversary can cause unintended behavior by performing slight modifications to the input at test-time. In neural networks used as classifiers, these *adversarial examples* are produced by taking some normal instance that is classified correctly, and applying a slight perturbation to cause it to be misclassified (or even misclassified as a specific target label chosen by the adversary). This phenomenon, which has been shown to affect all state-of-the-art networks, poses a significant hindrance to deploying neural networks in safety-critical settings.

Many effective techniques have been proposed for generating adversarial examples Szegedy et al. (2014); Goodfellow et al. (2014); Moosavi-Dezfooli et al. (2015); Carlini & Wagner (2017a); Tramèr et al. (2017); and, conversely, several techniques have been proposed for training networks that are more robust to these examples Huang et al. (2015); Zheng et al. (2016); Hendrik Metzen et al. (2017); Hendrycks & Gimpel (2017); Madry et al. (2017); Tramèr et al. (2017). Unfortunately, it has proven difficult to accurately assess the robustness of any given defense by evaluating it against existing techniques for generating adversarial examples. In several cases, a defensive technique that was at first thought to produce robust networks was later shown to be susceptible to new kinds of attacks Carlini & Wagner (2017b). This ongoing cycle thus cast a doubt on any newly-proposed defensive technique.

In recent years, new techniques have been proposed for the *formal verification* of neural networks Pulina & Tacchella (2010; 2012); Katz et al. (2017b); Huang et al. (2016); Ehlers (2017). These techniques take a network and a desired property, and formally prove that the network satisfies the property — or provide an input for which the property is violated, if such an input exists. Specifically, for a given input point and some allowed amount of distortion under a given metric, verification can be used for finding an adversarial example or for soundly proving that no such examples exist within the allowed distortion. While verification tends to be significantly slower in finding adversarial examples than the aforementioned heuristic-based techniques Pulina & Tacchella (2012); Katz et al. (2017a;b), it can provide the much-needed rigor for assessing the adversarial robustness of neural networks.

In this paper we propose a method for using formal verification to assess the effectiveness of techniques for producing adversarial examples or defending against them. The key idea is to examine networks and apply verification to identify *ground-truth* adversarial examples. Formally, given a neural network $F$, a distance metric $d$, and an input $x$, we say that another input $x'$ is a ground-truth adversarial example for $x$ if it is the nearest point (with respect to the metric $d$) such that $F$ assigns different labels to $x$ and $x'$. It follows that all points whose distance to $x$ is smaller than the distance between $x$ and $x'$ are assigned the same label as $x$. The distance to the ground truth is thus an indication of how robust the network is to adversarial attacks at point $x$. Ground truths can serve multiple purposes: (i) if ground-truth adversarial examples are known for a set of points drawn from some meaningful distribution thought to represent real-world inputs, they can serve to estimate the robustness of a network as a whole, against any possible attack; (ii) they can be used for assessing attack techniques, by measuring the proximity of the adversarial examples that these attacks produce to the ground truths; and (iii) they can be used for assessing the effectiveness of defense techniques, by computing new ground truths for the hardened network and comparing them to the ground truths of the original network.

Our contributions can thus be summarized as follows:

- We suggest to use *ground-truth adversarial examples*, the provably closest adversarial with respect to some distance metric, as a tool for studying attacks and defenses.

- We find that first-order attack algorithms often produce near-optimal results, i.e. results that are close to a ground-truth adversarial example.

- We study adversarial training and find that it *does* increase robustness and does not overfit to a specific attack, as long as the attack is iterative.

The rest of this paper is organized as follows. In Section 2 we provide some necessary background. We then describe the experiments that we conducted in Section 3, and analyze their results in Section 4. Finally, we conclude with Section 5.

## 2 BACKGROUND AND NOTATION

**Neural network notation.** We regard a neural network as a function $F(\cdot)$ consisting of multiple layers $F = F_n \circ F_{n-1} \circ \cdots \circ F_1 \circ F_0$. In this paper we exclusively study feed-forward neural networks used for classification, where $F_n$ is the softmax activation function. We refer to the output of the second-to-last-layer of the network (the input to $F_n$) as the logits and denote this as $Z = F_{n-1} \circ \cdots \circ F_1 \circ F_0$. We define $\ell_F(x, y)$ to be the cross-entropy loss of the network $F$ on instance $x$ with true label $y$.

We focus here on networks for classifying greyscale MNIST LeCun et al. (1998) images. Input images with width $W$ and height $H$ are represented as points in the space $[0, 1]^{W \cdot H}$.

**Adversarial examples.** Szegedy et al. (2014) Given an input $x$, classified originally as target $t = F(x)$, and a new desired target $t' \neq t$, we call $x'$ a *targeted adversarial example* if $F(x') = t'$ and $x'$ is close to $x$ under some given distance metric. In this paper we focus on the $L_\infty$ and $L_1$ distance metrics. In the *non-targeted* case, $t'$ is not provided and the only requirement is that $F(x') \neq F(x)$.

**Generating adversarial examples.** We consider three popular methods for constructing adversarial examples:

1. The *Fast Gradient Method* (*FGM*) Goodfellow et al. (2014) is a one-step algorithm that takes a single step in the direction of the gradient.

$$x' = \text{FGM}(x) = \text{clip}_{[0,1]}(x + \epsilon \text{sign}(\nabla \ell_F(x, y)))$$

where $\epsilon$ controls the step size taken, and clip ensures that the adversarial example resides in the valid image space from 0 to 1.

2. The *Basic Iterative Method* (*BIM*) Kurakin et al. (2016) (sometimes also called *Projected Gradient Descent* Madry et al. (2017)) can be regarded as an iterative application of the fast gradient method. Initially it lets $x'_0 = x$ and then uses the update rule

$$x'_{i+1} = \text{clip}_{[x-\alpha, x+\alpha]}(FGM(x'_i))$$

Intuitively, in each iteration this attack takes a step of size $\epsilon$ as per the FGM method, but it iterates this process while keeping each $x'_i$ within the $\alpha$-sized ball of $x$.

3. The *Carlini and Wagner* (*CW*) Carlini & Wagner (2017a) method is an iterative attack that constructs adversarial examples by approximately solving the minimization problem $\min d(x, x')$ such that $F(x') = t'$ for the attacker-chosen target $t'$, where $d(\cdot)$ is an appropriate distance metric. Since the constrained optimization is difficult, instead they choose to solve $\min d(x, x') + c \cdot g(x')$ where $g(x')$ is a loss function that encodes how close $x'$ is to being adversarial. Specifically, they set

$$g(x') = \max(\max\{Z(x')_i : i \neq t'\} - Z(x')_{t'}, 0).$$

$Z(\cdot)$, the logits of the network, are used instead of the softmax output because it was found to provide superior results. Although it was originally constructed to optimize for $L_2$ and $L_\infty$ distortion, we add the ability to use it with $L_1$ distortion in this paper.

**Neural network verification.** Neural networks excel at generalization: they are trained over a small set of inputs, and can then perform well, in general, on previously-unseen inputs. However, the intended use of deep neural networks as controllers in safety-critical systems Julian et al. (2016); Bojarski et al. (2016), on which human lives will depend, has created an interest in *formally proving* that a network behaves as expected for *any possible input*. In software and hardware systems, the rigorous checking that a system satisfies a prescribed property for any possible input is referred to as verification; and in recent years, several techniques have emerged for verifying deep neural networks Pulina & Tacchella (2010; 2012); Huang et al. (2016); Ehlers (2017); Katz et al. (2017b).

Here we focus on the recently-proposed Reluplex algorithm Katz et al. (2017b): an approach that can effectively tackle networks with piecewise-linear activation functions, such as rectified linear units (ReLUs) or max-pooling layers. Reluplex takes as input a network $F$ and a property $\varphi$, given as a convex set of linear constraints on the network's inputs and outputs, and checks whether $F$ satisfies $\varphi$ (this problem is NP-complete Katz et al. (2017b)). Reluplex is a simplex-based SMT solver, and it operates by repeatedly finding inputs that satisfy the network's linear constraints (the weighted sums and the constraints in $\varphi$) and then adjusting them so that they also satisfy the network's non-linear constraints (the activation functions). Reluplex is known to be sound and complete, making it suitable for establishing ground truths.

In Katz et al. (2017b) it is shown that Reluplex can be used to determine whether there exists an adversarial example within distance $\delta$ of some input point $x$. This is performed by encoding the neural network itself and the constraints regarding $\delta$ as a set of linear equations and ReLU constraints, and then having Reluplex attempt to prove the property that "there does not exist an input point within distance $\delta$ of $x$ that is assigned a different label than $x$". Reluplex either responds that the property holds, in which case there is no adversarial example within distance $\delta$ of $x$, or it returns a counter-example which constitutes the sought-after adversarial input (a similar query can test the existence of targeted adversarial examples). By invoking Reluplex iteratively and applying binary search, one can approximate the optimal $\delta$ (i.e., the largest $\delta$ for which no adversarial example exists) up to a desired precision Katz et al. (2017b).

The proof-of-concept implementation of Reluplex described in Katz et al. (2017b) supported only networks with the ReLU activation function, and could only handle the $L_\infty$ norm as a distance metric. Here we use a simple encoding that allows us to use it for the $L_1$ norm as well.

**Adversarial training.** Adversarial training Szegedy et al. (2014) is perhaps the first proposed defense against adversarial examples, and is a conceptually straightforward approach. The defender trains a classifier, generates adversarial examples for that classifier, retrains the classifier using the adversarial examples, and repeats.

Recent work has shown Madry et al. (2017) that for networks with sufficient capacity, adversarial training can be an effective defense even against the most powerful attacks today. However, it has been shown that performing adversarial training with a weaker attack, such as the fast gradient method, does not increase robustness substantially against stronger attacks. It is an open question whether adversarial training using stronger attacks actually increases robustness against arbitrary adversarial examples, or whether such training would only be effective against known attacks but would be vulnerable against future, stronger attacks. In our evaluation below, we find that adversarially training a small network using strong iterative attacks does indeed increase its adversarial robustness.

## 3 MODEL SETUP

**Standard MNIST model.** The problem of neural network verification that we consider here is an NP-complete problem Katz et al. (2017b), and despite recent progress only networks with a few hundred nodes can be soundly verified. Thus, in order to evaluate our approach we trained a small network over the MNIST data set. This network is a fully-connected, 3-layer network that achieves a 97% accuracy despite having only 20k weights and consisting of fewer than 100 hidden neurons (24 in each layer). As verification of neural networks becomes more scalable in the future, our approach could become applicable to larger networks and additional data sets.

**Adversarially trained MNIST model.** We construct our adversarially trained network as described in Madry et al. (2017). To summarize the process, we perform training in the usual manner, with the following modification: given the current model $F_\theta$, after a given minibatch $\{x_i, y_i\}$ is selected, we construct a new minibatch $\{x_i', y_i\}$ so that $x_i'$ is an adversarial example. Specifically, we set $x_i' = x_i + \delta_i$ where $\|\delta_i\|_\infty \le \epsilon$, and $F_\theta(x_i') \ne y_i$. Madry et al. (2017) use the basic iterative method described in Section 2 to generate these adversarial examples.

We applied this adversarial training to our small network described above. Because our network capacity is smaller than that of the original network used by Madry et al. (2017), we set $\epsilon$ to 0.15 instead of the original 0.3.

**Reluplex implementation.** For verification, we use the proof-of-concept implementation of Reluplex available online Katz et al. (2017c). The only non-linear operator that this implementation was originally designed to support is the ReLU function, but we observe that it can support also $\max$ operators using the following encoding:

$$\max(x, y) = \text{ReLU}(x - y) + y$$

This allows the encoding of absolute values as well:

$$|x| = \max(x, -x) = \text{ReLU}(2x) - x$$

Because the $L_1$ distance between two points is defined as a sum of absolute values, using this encoding we were able to use the tool from Katz et al. (2017b) to measure distances with the $L_1$ norm in addition to the $L_\infty$ norm, without modifying the tool's code. (Note that this encoding can be used to encode max-pooling layers into Reluplex, although we did not experiment with such layers in this paper.) We point out, however, that an increase in the number of ReLU constraints in the input adversely affects Reluplex's performance. For example, in the case of the MNIST dataset, encoding an $L_1$ distance constraint entails adding a ReLU operator for each of the 784 input coordinates; and indeed, as we show below, experiments using $L_1$ typically took longer to finish than those using $L_\infty$.

**Finding ground truths.** Each individual experiment that we conducted included a network $F$, a distance metric $d \in \{L_1, L_\infty\}$, an input point $x$, a target label $t' \ne F(x)$, and an initial input $x'_{init}$ for which $F(x'_{init}) = t'$. The goal of the experiment was then to find a ground-truth example $x_{t'}$, such that $F(x_{t'}) = t'$ and $d(x, x_{t'})$ is minimal. As explained in Section 2, this is performed by iteratively invoking Reluplex and performing a binary search. This procedure is given as Algorithm 1.

---

**Algorithm 1** Find Ground Truth $(F, d, x, t', x'_{init})$

---

1: $\delta_{min} := 0$
2: $\delta_{max} := \|x - x'_{init}\|_d$
3: $x'_{best} := x'_{init}$
4: **while** $\delta_{max} - \delta_{min} > 10^{-4}$ **do**
5:     $\delta := (\delta_{max} + \delta_{min})/2$
6:     Invoke Reluplex to test whether $\exists x'. \|x - x'\|_d \leq \delta \wedge F(x') = t'$
7:     **if** $x'$ exists **then**
8:         $\delta_{max} := \|x - x'\|_d$
9:         $x'_{best} := x'$
10:    **else**
11:        $\delta_{min} := \delta$
12: **return** $\delta_{max}, x'_{best}$

---

Intuitively, $\delta_{max}$ indicates the distance to the closest adversarial input currently known, and the distance to the ground-truth input is known to be in the range between $\delta_{min}$ and $\delta_{max}$. Thus, $\delta_{max}$ is initialized using the distance of the initial example provided via $x'_{init}$, and $\delta_{min}$ is initialized to 0. The search procedure iteratively shrinks the range $\delta_{max} - \delta_{min}$ until it is below a certain threshold (we used $10^{-4}$ for our experiments). It then returns $\delta_{max}$ as the distance to the ground truth, and this is guaranteed to be accurate up to the specified precision. The ground-truth input itself is also returned.

For the initial $x'_{init}$ in our experiments we used an adversarial input found using the CW attack. This was done to improve performance: while the experiment would work with any point $x'_{init}$ that is labeled $t'$, Reluplex invocations are computationally expensive, and so it is better to start with an $x'_{init}$ that is as close as possible to $x$ in order to reduce the number of required iterations until $\delta_{max} - \delta_{min}$ is sufficiently small. This was another reason due to which experiments using the $L_1$ distance metric were slower than those using $L_\infty$ (in addition to the higher number of ReLU constraints): the initial distances when using $L_1$ were typically much larger, which required a higher number of iterations.

## 4 EVALUATION

For evaluation purposes we arbitrarily selected 10 source images with known labels from the MNIST test set. We considered two neural networks — the one described in Section 3, denoted $N$, and also a version of $N$ that had been trained with adversarial training as described in Section 3, denoted $\bar{N}$. We also considered two distance metrics, $L_1$ and $L_\infty$. For every combination of neural network, distance metric and labeled source image $x$, we considered each of the 9 other possible labels for $x$. For each of these we used the CW attack to produce an initial targeted adversarial example, and then used Algorithm 1 to search for a ground-truth example. An analysis of the results is given in Table 1, and a graphical depiction of some of results appears in Fig. 1. A more extensive depiction appears in the appendix at the end of the paper.

Each major row of Table 1 corresponds to a specific neural network and distance metric (as indicated in the first column), and describes 90 individual experiments (10 inputs, times 9 target labels for each input). The first sub-row within each row considers just those experiments for which Algorithm 1 terminated successfully, whereas the second sub-row considers all 90 examples, including those where Algorithm 1 timed out. Whenever a timeout occurred, we considered the last (smallest) $\delta_{max}$ that was discovered by the search before it timed out as the algorithm's output. The other columns of the table indicate the average distance to the adversarial examples found by the CW attack, the average distance to the ground-truth adversarial examples found by our technique, and the average improvement rate of our technique over the CW attack.

Performance-wise, our experiments displayed a high degree of variability. The number of iterations required by Algorithm 1 ranged from 1 to 19, and was affected by the test point, the target label, the network, the distance metric and the level of precision that we used ($10^{-4}$). The time each iteration took also varied significantly, between a few seconds and several days, with a median time of 2.3 hours.

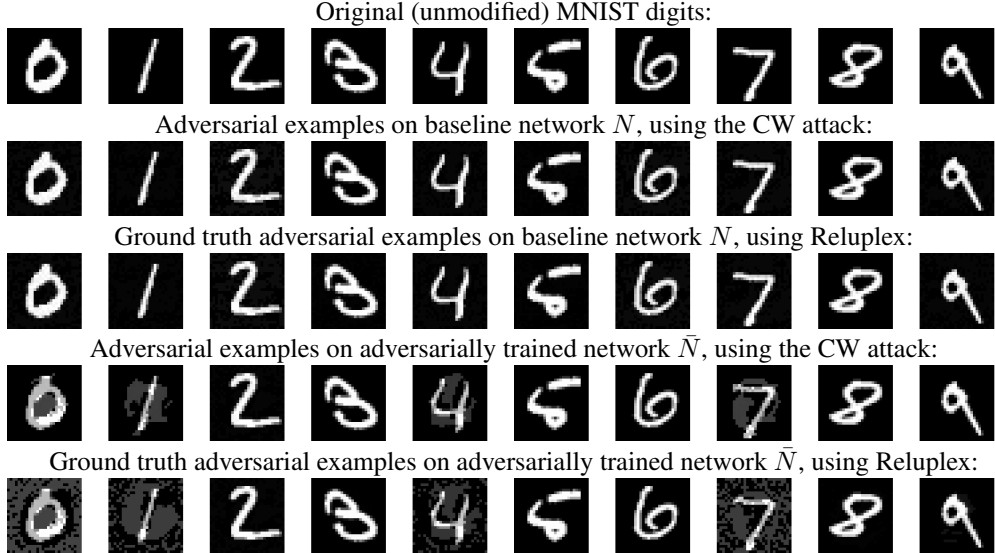

Figure 1: Our 10 test points, and non-targeted adversarial inputs for these points in our various test settings.

Table 1: Evaluating our technique on the MNIST dataset

|  | #Points | CW | Ground Truth | % Improvement |
|---|---|---|---|---|
| $N, L_\infty$ | 39/90 | 0.042 | 0.039 | 11.346 |
|  | 90/90 | 0.063 | 0.061 | 6.402 |
| $N, L_1$ | 12/90 | 2.919 | 2.69 | 19.947 |
|  | 90/90 | 7.551 | 7.488 | 3.375 |
| $\bar{N}, L_\infty$ | 83/90 | 0.215 | 0.197 | 11.423 |
|  | 90/90 | 0.219 | 0.206 | 10.587 |
| $\bar{N}, L_1$ | 71/90 | 7.205 | 7.13 | 5.691 |
|  | 90/90 | 8.187 | 8.128 | 4.491 |

Below we analyze the results in order to draw conclusions regarding the CW attack and the defense of Madry et al. (2017). While these results naturally hold only for the networks we used and the inputs we tested, we believe they provide some intuition as to how well the tested attack and defense techniques perform in general. We intend to make our data publicly available, and we encourage others to (i) evaluate new attack techniques using the ground-truth examples we have already discovered; and (ii) use this approach for evaluating new defensive techniques.

## 4.1 EVALUATING ATTACKS

**Iterative attacks produce near-optimal adversarial examples.** As shown by Table 1, the adversarial examples produced by the CW attack are, on average, within $11.4\%$ of the ground truth when using the $L_\infty$ norm, and within $5.7\%$ of the ground truth when using $L_1$ (we consider here just the terminated experiments, and ignore the $N, L_1$ category where too few experiments terminated to draw a meaningful conclusion).

Additional experiments that we ran (not displayed in Table 1) indicated that the iterative CW attack performs substantially better than the single-step FGM method. On the $N$ network with $L_\infty$ distance metric, with $\epsilon$ set to 0.12 which is twice the average distance to the ground truth, FGM was only able to generate a targeted adversarial example in $39\%$ of the cases. This is to be expected: FGM was designed to show the linearity of neural networks, not to produce high-quality adversarial examples.

Table 2: Comparing the 37 instances on which Algorithm 1 terminated for both $N, L_\infty$ and $\bar{N}, L_\infty$.

|  | #Points | CW | Ground Truth | % Improvement |
|---|---|---|---|---|
| $N, L_\infty$ | 37/37 | 0.043 | 0.04 | 11.814 |
| $\bar{N}, L_\infty$ | 37/37 | 0.185 | 0.171 | 10.624 |

**There is still room for improving iterative attacks.** Even on this very small and simple neural network, we observed that in many instances the ground-truth adversarial example has a $30\%$ or $40\%$ lower distortion rate than the best iterative adversarial example.

The cause for this is simple: gradient descent only finds a local minimum, not a global minimum. We have found that if we take a small step from the original image in the direction of the ground truth, then gradient descent will converge to the ground truth much more often. This is in line with the suggestion raised in Carlini & Wagner (2017a) and Tramèr et al. (2017), that it is useful to take a random step before searching for the nearest adversarial example.

**Suboptimal results are correlated.** We have found that whenever the iterative attack performs suboptimally compared to the ground truths for a specific input and target label, it will often perform poorly for that input and many other target labels as well. These instances are not always of a large absolute distortion — but rather, a large relative gap on one instance often indicates that the relative gap will be large for other target labels. For instance, on the adversarially trained network attacked under $L_\infty$ distance, all ground-truth adversarial examples for one of the instances (the image of a "9" that we used) were between $21\%$ and $47\%$ better than the iterative attack results.

When we examined the most extreme cases in which this phenomenon was observed, we found that, similarly to the case described above, the large gap was caused by gradient descent initially leading *away* from the ground truth for most targets, resulting in the discovery of an inferior, local minimum.

### 4.2 EVALUATING DEFENSES

For the purpose of evaluating the defensive technique of Madry et al. (2017), we compared the $N, L_\infty$ and $\bar{N}, L_\infty$ experiments (the $L_1$ experiments were disregarded because of the small number of experiments that terminated for the $N, L_1$ case). Specifically, we compared the $N, L_\infty$ and $\bar{N}, L_\infty$ experiments on the subset of 37 instances that terminated for both experiments. The results are analyzed in Table 2.

**The defense of Madry et al. (2017) is effective.** Our evaluation suggests that the Madry defense is indeed effective: it improves the distance to the ground truth by an average of 427% (from an average of 0.04 to an average of 0.171) on our small network.

Another interesting observation is that while the Madry defense improves the overall situation, we found several points for which it actually made things worse — i.e., the ground truth for the hardened network was smaller than that of the original network. This behavior was observed for 7 out of the 37 aforementioned experiments, with the average percent of degradation being 12.8%. This seems to highlight the necessity of evaluating the effectiveness of a defensive technique, and the robustness of a network in general, over a large dataset of points. The question of how to pick a "good" set of points that would adequately represent the behavior of the network remains open; one possible direction is to use a clustering-based approach Gopinath et al. (2017).

**Training on iterative attacks does not overfit.** Overfitting is an issue that is often encountered when performing adversarial training, meaning that the hardened network becomes specifically tailored for the type of attack used during training. When this occurs, the hardened network will have high accuracy against that specific attack, but low accuracy on other attacks. We have found no evidence of overfitting when performing the adversarial training of Madry et al. (2017): the ground truths improve on the CW attack by approximately $11.4\%$ on both the hardened and untrained networks. This indicates that the Madry network did not become more susceptible to the CW attack as a result of its hardening using a different technique.

**The defense of Madry et al. (2017) makes networks easier to formally analyze.** For both the $L_\infty$ and $L_1$ distance metrics, it seems significantly easier to analyze the robustness of the adversarially trained network: when using $L_\infty$, Algorithm 1 terminated on 83 of the 90 instances on the adversarially trained network, versus 39 on the standard network; and for $L_1$, the termination rate was 71 for the hardened network compared to just 12 on the standard network. We are still looking into the reason for this behavior. Naively, one might assume that, because the initial adversarial examples $x'_{init}$ provided to Algorithm 1 are farther away in the hardened network, these experiments would take longer to converge — but we see an opposite behavior.

One possible explanation could be that the adversarially trained network makes less use of the non-linear ReLU units, and is therefore more amenable to analysis with Reluplex. We empirically checked that this was not the case. For a given instance, we tracked, for each ReLU unit in the network, whether it was in the saturated zero region, or the linear $x = y$ region. We then computed the nonlinearity of the network as the number of units that change from the saturated region to the linear region, or vice versa, when going from the given input to the discovered adversarial example. We found that there was no statistically significant difference between the nonlinearity of the two networks.

## 5 CONCLUSION

Neural networks hold great potential as controllers in safety-critical systems, but their susceptibility to adversarial examples poses a significant hindrance. The development of defensive techniques is difficult when they are measured only against existing attacks. The burgeoning field of neural network verification can mitigate this problem, by allowing us to obtain an absolute measurement of the usefulness of a defense, regardless of the attack to be used against it.

In this paper, we introduce ground-truth adversarial examples and show how to construct them with formal verification approaches. We evaluate one recent attack Carlini & Wagner (2017a) and find it often produces adversarial examples whose distance is within $5.9\%$ to $12.9\%$ of optimal, and one defense Madry et al. (2017), and find that it increases distortion to the nearest adversarial example by an average of $427\%$ on the MNIST dataset for our tested networks. To the best of our knowledge, this is the first proof of robustness increase for a defense.

Currently available verification tools afford limited scalability, which means experiments can only be conducted on small networks. However, as better verification techniques are developed, this limitation is expected to be mitigated. Orthogonally, when preparing to use a neural network in a safety-critical setting, users may choose to design their networks as to make them particularly amenable to verification techniques — e.g., by using specific activation functions or network topologies — so that strong guarantees about their correctness and robustness may be obtained.

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

# Appendix

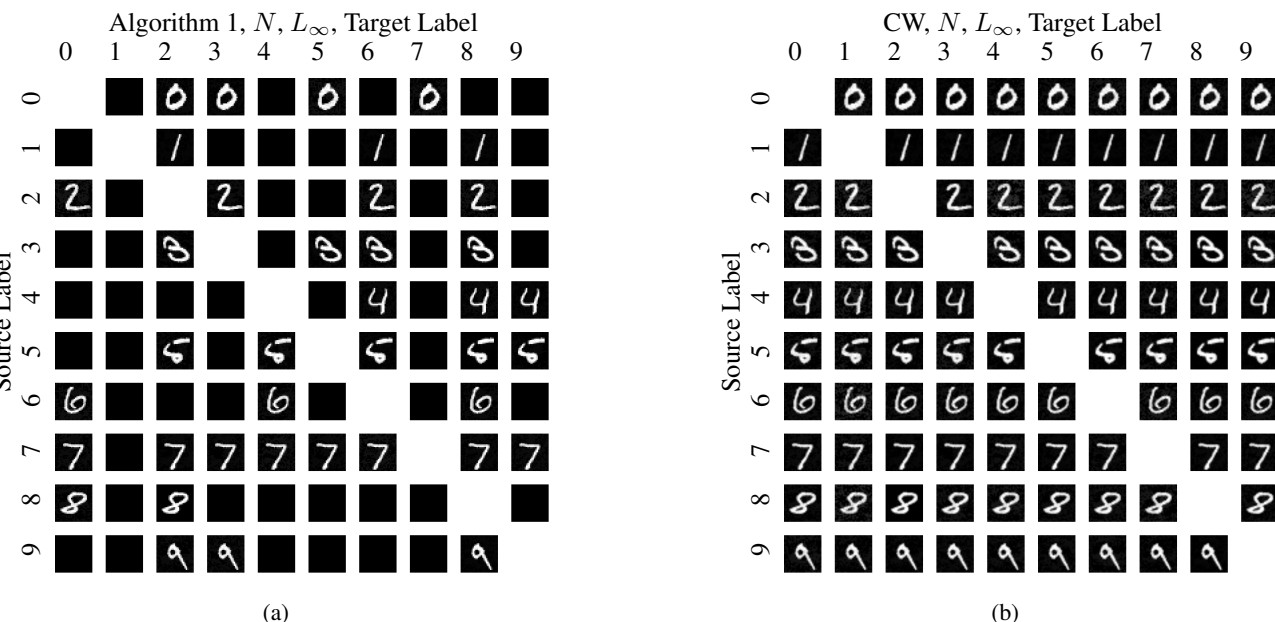

(a)

(b)

Figure 2: Adversarial examples constructed on a standard MNIST classifier under the $L_\infty$ distance metric using (left) Reluplex and (right) Carlini and Wagner's attack.

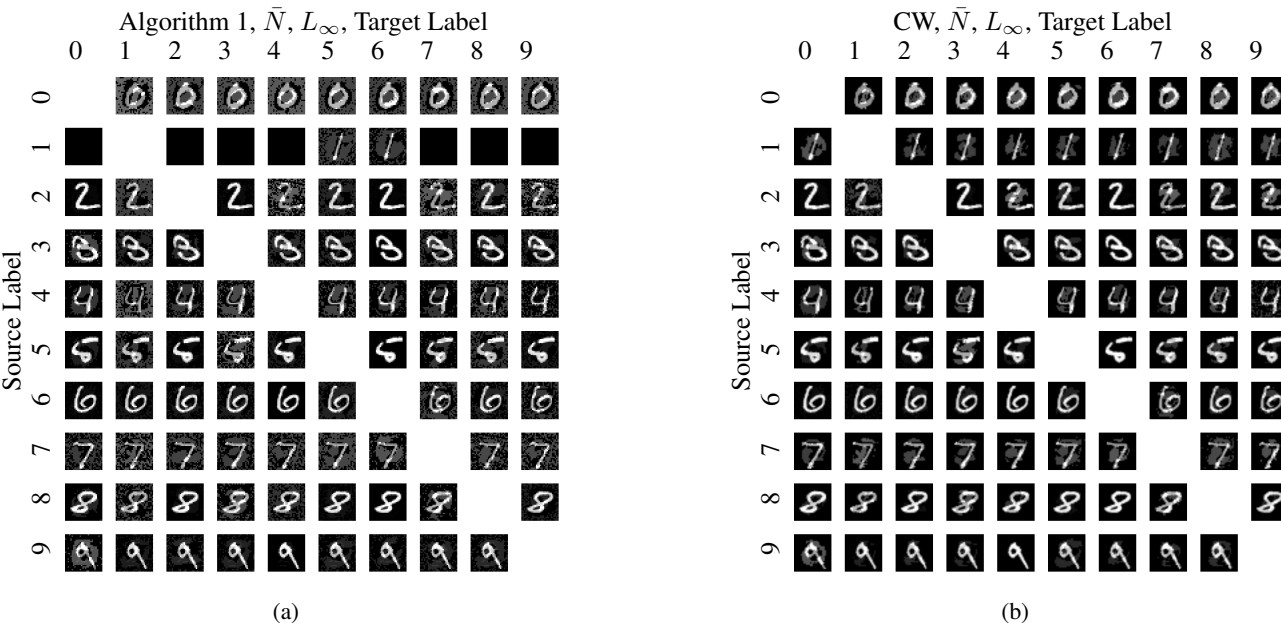

(a)

(b)

Figure 3: Adversarial examples constructed on a adversarially trained MNIST classifier under the $L_\infty$ distance metric using (left) Reluplex and (right) Carlini and Wagner's attack.

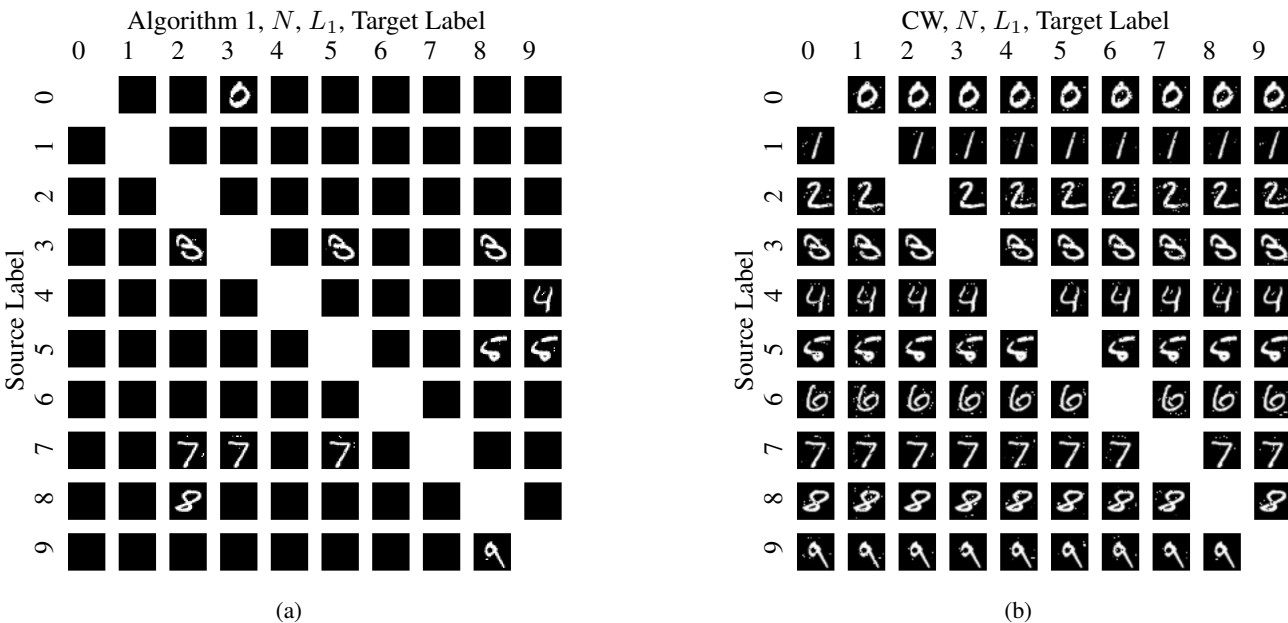

Figure 4: Adversarial examples constructed on a standard MNIST classifier under the $L_1$ distance metric using (left) Reluplex and (right) Carlini and Wagner's attack.

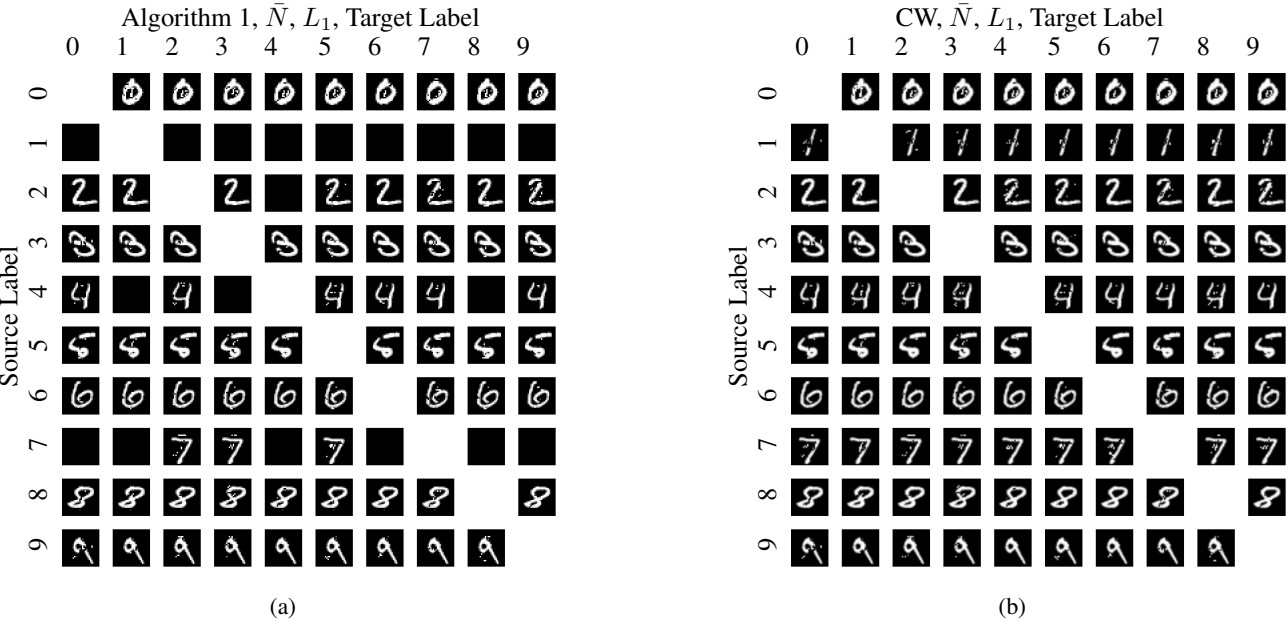

Figure 5: Adversarial examples constructed on a adversarially trained MNIST classifier under the $L_1$ distance metric using (left) Reluplex and (right) Carlini and Wagner's attack.

