# OpenReview forum: "Ground-Truth Adversarial Examples"
_ICLR.cc/2018/Conference — Reject_

### Official Review · AnonReviewer2 · 2017-11-27
**Theoretically interesting but practically maybe limited**

**Rating:** 5
**Confidence:** 4

**Review:**

Summary: The paper proposes a method to compute adversarial examples with minimum distance to the original inputs, and to use the method to do two things: Show how well heuristic methods do in finding "optimal/minimal" adversarial examples (how close the come to the minimal change that flips the label) and to assess how a method that is designed to make the model more robust to adversarial examples actually works.

Pros:

I like the idea and the proposed applications. It is certainly highly relevant, both in terms of assessing models for critical use cases as well as a tool to better understand the phenomenon.

Some of the suggested insights in the analysis of defense techniques are interesting.

Cons:

The is not much technical novelty. The method boils down to applying Reluplex (Katz et al. 2017b) in a binary search (although I acknowledge the extension to L1 as distance metric).

The practical application of the method is very limited since the search is very slow and is only feasible at all for relatively small models. State-of-the-art practical models that achieve accuracy rates that make them interesting for deployment in potentially safety critical applications are out of reach for this analysis. The network analysed here does not reach the state-of-the-art on MNIST from almost two decades ago. The analysis also has to be done for each sample. The long runtime does not permit to analyse large amounts of input samples, which makes the analysis in terms of the increase in robustness rather weak. The statement can only be made for the very limited set of tested samples.

It is also unclear whether it is possible to include distance metrics that capture more sophisticated attacks that fool network even under various transformations of the input.
The paper does not consider the more recent and highly relevant Moosavi-Dezfooli et al. “Universal Adversarial Perturbations” CVPR 2017.

The distance metrics that are considered are only L_inf and L1, whereas it would be interesting to see more relevant “perceptual losses” such as those used in style transfer and domain adaptation with GANs.

Minor details:
* I would consider calling them “minimal adversarial samples” instead of “ground-truth”.
* I don’t know if the notation in the Equation in the paragraph describing Carlini & Wagner comes from the original paper, but the inner max would be easier to read as \max_{i \neq t} \{Z(x’)_i \}
* Page 3 “Neural network verification”: I dont agree with the statement that neural networks commonly are trained on “a small set of inputs”.
* Algorithm 1 is essentially only a description of binary search, which should not be necessary.
* What is the timeout for the computation, mentioned in Sec 4?
* Page 7, second paragraph: I wouldn’t say the observation is in line with Carlini & Wagner, because they take a random step, not necessarily one in the direction of the optimum? That’s also the conclusion two paragraphs below, no?
* I don’t fully agree with the conclusion that the defense of Madry does not overfit to the specific method of creating adversarial examples. Those were not created with the CW attack, but are related because CW was used to initialize the search.

---

### Official Review · AnonReviewer3 · 2017-11-27
**Interesting but not too convincing**

**Rating:** 4
**Confidence:** 4

**Review:**

The authors propose to employ provably minimal-distance examples as a tool to evaluate the robustness of a trained network. This is demonstrated on a small-scale network using the MNIST data set.

First of all, I find it striking that a trained network with 97% accuracy (as claimed by the authors) seems extremely brittle -- considering the fact that all the adversarial examples in Figure 1 are hardly borderline examples at all, at least to my eyes. This does reinforce the (well-known?) weakness of neural networks in general. I therefore find the authors' statement on page 3 disturbing: "... they are trained over a small set of inputs, and can then perform well, in general, on previously-unseen inputs" -- which seems false (with high probability over all possible worlds).

Secondly, the term "ground truth" example seems very misleading to me. Perhaps "closest misclassified examples"?

Finally, while the idea of "closest misclassified examples" seems interesting, I am not convinced that they are the right way to go when it comes to both building and evaluating robustness. All such examples shown in the paper are indeed within-class examples that are misclassified. But we could equally consider another extreme, where the trained network is "over-regularized" in the sense that the closest misclassified examples are indeed from another class, and therefore "correctly" misclassified. Adding these as adversarial examples could seriously degrade the accuracy.

Also, for building robustness, one could argue that adding misclassified examples that are "furthest" (i.e. closest to the true decision boundary) is a much more efficient training approach, since a few of these can possibly subsume a large number of close examples.

---

### Official Review · AnonReviewer1 · 2017-11-28
**Novel idea, but more experiments needed to support findings**

**Rating:** 6
**Confidence:** 3

**Review:**

The paper describes a method for generating so called ground truth adversarial examples: adversaries that have minimal (L1 or L_inf) distance to the training example used to generate them. The technique uses the recently developed reluplex, which can be used to verify certian properties of deep neural networks that use ReLU activations. The authors show how the L1 distance can be formulated using a ReLU and therefore extend the reluplex also work with L1 distances. The experiments on MNIST suggest that the C&W attack produces close to optimal adversarial examples, although it is not clear if these findings would transfer to larger more complex networks. The evaluation also suggests that training with iterative adversarial examples does not overfit and does indeed harden the network to attacks in many cases.

In general, this is a nice idea, but it seems like the inherent computational cost will limit the applicability of this approach to small networks and datasets for the time being. Incidentally, it would have been useful if the authors provided indicative information on the computational cost (e.g. in the form of time on a standard GPU) for generating these ground truths and carrying out experiments.

The experiments are quite small scale, which I expect is due to the computational cost of generating the adversarial examples. It is difficult to say how far the findings can be generalized from MNIST to more realistic situations. Tests on another dataset would have been welcomed.

Also, while interesting, are adversarial examples that have minimal L_p distance from training examples really that useful in practice? Of course, it's nice that we can find these, but it could be argued that L_p norms are not a good way of judging the similarity of an adversarial example to a true example. I think it would be more useful to investigate attacks that are perceptually insignificant, or attacks that operate in the physical world, as these are more likely to be a concern for real world systems.

In summary, while I think the paper is interesting, I suspect that the applicability of this technique is possibly limited at present, and I'm unsure how much we can really read into the findings of the paper when the experiments are based on MNIST alone.

---

### Decision · Program_Chairs · 2018-01-29
**ICLR 2018 Conference Acceptance Decision**

**Decision:**

Reject

**Comment:**

This paper describes a method to generate provably 'optimal' adversarial examples, leveraging the so-called 'Reluplex' technique, which can evaluate properties of piece-wise linear representations.
Reviewers agreed that incorporating optimality certificates into adversarial examples is a promising direction to follow, but were also concerned about the lack of empirical justification the current paper provides and missed discussion about the relevance of choosing Lp distances. They all recommended pushing experiments to more challenging datasets before the paper can be accepted, and the AC shares the same advice.